# FEATURE QUANTIZATION FOR PARSIMONIOUS AND INTERPRETABLE PREDICTIVE MODELS

## ABSTRACT

For regulatory and interpretability reasons, the logistic regression is still widely used by financial institutions to learn the refunding probability of a loan from applicant's historical data. To improve prediction accuracy and interpretability, a preprocessing step quantizing both continuous and categorical data is usually performed: continuous features are discretized by assigning factor levels to intervals and, if numerous, levels of categorical features are grouped. However, a better predictive accuracy can be reached by embedding this quantization estimation step directly into the predictive estimation step itself. By doing so, the predictive loss has to be optimized on a huge and untractable discontinuous quantization set. To overcome this difficulty, we introduce a specific two-step optimization strategy: first, the optimization problem is relaxed by approximating discontinuous quantization functions by smooth functions; second, the resulting relaxed optimization problem is solved *via* a particular neural network and stochastic gradient descent. The strategy gives then access to good candidates for the original optimization problem after a straightforward *maximum a posteriori* procedure to obtain cutpoints. The good performances of this approach, which we call *glmdisc*, are illustrated on simulated and real data from the UCI library and Crédit Agricole Consumer Finance (a major European historic player in the consumer credit market). The results show that practitioners finally have an automatic all-in-one tool that answers their recurring needs of quantization for predictive tasks.

## 1 MOTIVATION

As stated by Hosmer Jr et al. (2013), in many application contexts (credit scoring, biostatistics, *etc.*), logistic regression is widely used for its simplicity, decent performance and interpretability in predicting a binary outcome given predictors of different types (categorical, continuous). However, to achieve higher interpretability, continuous predictors are sometimes discretized so as to produce a "scorecard", *i.e.* a table assigning a grade to an applicant in credit scoring (or a patient in biostatistics, *etc.*) depending on its predictors being in a given interval. Discretization is also an opportunity for reducing the (possibly large) modeling bias which can appear in logistic regression as a result of the linearity assumption on the continuous predictors in the model. Indeed, this restriction can be overcome by approximating the true predictive mapping with a step function where the tuning of the steps and their sizes allows more flexibility. However, the resulting increase of the number of parameters can lead to an increase in variance (overfitting) as shown in Yang & Webb (2009). Thus, a precise tuning of the discretization procedure is required. Likewise when dealing with categorical features which take numerous levels, their respective regression coefficients suffer from high variance. A straightforward solution formalized by Maj-Kańska et al. (2015) is to merge their factor levels which leads to less coefficients and therefore less variance.

From now on, the generic term quantization will stand for both discretization of continuous features and level grouping of categorical ones. Its aim is to improve the prediction accuracy. Such a quantization can be seen as a special case of *representation learning*, but suffers from a highly combinatorial optimization problem whatever the predictive criterion used

to select the best quantization. The present work proposes a strategy to overcome these combinatorial issues by invoking a relaxed alternative of the initial quantization problem leading to a simpler estimation problem since it can be easily optimized by a specific neural network. This relaxed version serves as a plausible quantization provider related to the initial criterion after a classical thresholding (*maximum a posteriori*) procedure.

The outline of this work is the following. In the next section, we formalize both continuous and categorical quantization. Selecting the best quantization in a predictive setting is reformulated as a model selection problem on a huge discrete space. In Section 3, a particular neural network architecture is used to optimize a relaxed version of this criterion and propose good quantization candidates. Section 4 is dedicated to numerical experiments on both simulated and real data from the field of Credit Scoring, highlightening the good results offered by the use of this new method without any human intervention. A final section concludes the work by stating also new challenges.

## 2 QUANTIZATION AS A COMBINATORIAL CHALLENGE

### 2.1 QUANTIZATION: DEFINITION

**General principle** The quantization procedure consists in turning a $d$-dimensional raw vector of continuous and/or categorical features $\boldsymbol{x} = (x_1, \ldots, x_d)$ into a $d$-dimensional categorical vector via a component wise mapping $\boldsymbol{q} = (\boldsymbol{q}_j)_1^d$:

$$\boldsymbol{q}(\boldsymbol{x}) = (\boldsymbol{q}_1(x_1), \ldots, \boldsymbol{q}_d(x_d)),$$

where each of the $\boldsymbol{q}_j$'s is a vector of $m_j$ dummies:

$$q_{j,h}(\cdot) = 1 \text{ if } x_j \in C_{j,h}, 0 \text{ otherwise}, 1 \leq h \leq m_j, \tag{1}$$

where $m_j$ is an integer and the sets $C_{j,h}$ are defined with respect to each feature type as we describe just below.

**Raw continuous features** If $x_j$ is a continuous component of $\boldsymbol{x}$, quantization $\boldsymbol{q}_j$ has to perform a discretization of $x_j$ and the $C_{j,h}$'s, $1 \leq h \leq m_j$, are contiguous intervals

$$C_{j,h} = (c_{j,h-1}, c_{j,h}] \tag{2}$$

where $c_{j,1}, \ldots, c_{j,m_j-1}$ are increasing numbers called cutpoints, $c_{j,0} = -\infty$, $c_{j,m_j} = \infty$.

For example, the quantization of the unit segment in thirds would be defined as $m_j = 3$, $c_{j,1} = 1/3$, $c_{j,2} = 2/3$ and subsequently $\boldsymbol{q}_j(0.1) = (1, 0, 0)$.

**Raw categorical features** If $x_j$ is a categorical component of $\boldsymbol{x}$, quantization $\boldsymbol{q}_j$ consists in grouping levels of $x_j$ and the $C_{j,h}$s form a partition of the set, say $\{1, \ldots, l_j\}$, of levels of $x_j$:

$$\bigsqcup_{h=1}^{m_j} C_{j,h} = \{1, \ldots, l_j\}.$$

For example, the grouping of levels encoded as "1" and "2" would yield $C_{j,1} = \{1, 2\}$ such that $\boldsymbol{q}_j(1) = \boldsymbol{q}_j(2) = (1, 0, \ldots, 0)$.

**Notations for the quantization family** In both continuous and categorical cases, keep in mind that $m_j$ is the dimension of $\boldsymbol{q}_j$. For notational convenience, the (global) order of the quantization $\boldsymbol{q}$ is set as

$$|\boldsymbol{q}| = \sum_{j=1}^{d} m_j.$$

The space where quantizations $\boldsymbol{q}$ live will be denoted by $\mathcal{Q}$ in the sequel.

**Literature review** The current practice of quantization is prior to any predictive task, thus ignoring its consequences on the final predictive ability. It consists in optimizing a heuristic criterion, often totally unrelated (unsupervised methods) or at least explicitly (supervised methods) to prediction, and mostly univariate (each feature is quantized irrespective of other features' values). The cardinality of the quantization space $\mathcal{Q}$ can be calculated explicitely w.r.t. $d$, $(m_j)_1^d$ and, for categorical features, $l_j$. It is huge, so that a greedy approach is intractable and such heuristics are needed. Many algorithms have thus been designed and a review of approximatively 200 discretization strategies, gathering both criteria and related algorithms, can be found in Ramírez-Gallego et al. (2016). For factor levels grouping, we found no such taxonomy, but some discretization methods, *e.g.* $\chi^2$ independence test-based methods can be naturally extended to this type of quantization, which is for example what the CHAID algorithm, proposed by Kass (1980) and applied to each categorical feature, relies on.

## 2.2 Quantization embedded in a predictive process

**Logistic regression on quantized data** Quantization is a widespread preprocessing step to perform a learning task consisting in predicting, say, a binary variable $y \in \{0, 1\}$, from a quantized predictor $\boldsymbol{q}(\boldsymbol{x})$, through, say, a parametric conditional distribution $p_{\boldsymbol{\theta}}(y|\boldsymbol{q}(\boldsymbol{x}))$ like logistic regression. Considering quantized data instead of raw data has a double benefit. First, the quantization order $|\boldsymbol{q}|$ acts as a tuning parameter for controlling the model's flexibility and thus the bias/variance trade-off of the estimate of the parameter $\boldsymbol{\theta}$ (or of its predictive accuracy) for a given dataset. This claim becomes clearer with the example of logistic regression we focus on, as a still very popular model for many practitioners. It is classically described by

$$\ln\left(\frac{p_{\boldsymbol{\theta}}(1|\boldsymbol{q}(\boldsymbol{x}))}{1 - p_{\boldsymbol{\theta}}(1|\boldsymbol{q}(\boldsymbol{x}))}\right) = \theta_0 + \sum_{j=1}^{d} \boldsymbol{\theta}_j' \cdot \boldsymbol{q}_j(x_j), \tag{3}$$

where $\boldsymbol{\theta} = (\theta_0, (\boldsymbol{\theta}_j)_1^d) \in \mathbb{R}^{|\boldsymbol{q}|+1}$ and $\boldsymbol{\theta}_j = (\theta_j^1, \ldots, \theta_j^{m_j})$ with $\theta_j^{m_j} = 0$, $j = 1 \ldots d$, for identifiability reasons. Second, at the practitioner level, the previous tuning of $|\boldsymbol{q}|$ through each feature's quantization order $m_j$, especially when it is quite low, allows an easier interpretation of the most important predictor values involved in the predictive process. Denoting the dataset by $(\mathbf{x}, \mathbf{y})$, with $\mathbf{x} = (\boldsymbol{x}_1, \ldots, \boldsymbol{x}_n)$ and $\mathbf{y} = (y_1, \ldots, y_n)$, the log-likelihood

$$\ell_{\boldsymbol{q}}(\boldsymbol{\theta}; (\mathbf{x}, \mathbf{y})) = \sum_{i=1}^{n} \ln p_{\boldsymbol{\theta}}(y_i|\boldsymbol{q}(\boldsymbol{x}_i)) \tag{4}$$

provides a Maximum Likelihood estimator $\hat{\boldsymbol{\theta}}_{\boldsymbol{q}}$ of $\boldsymbol{\theta}$ for a given quantization $\boldsymbol{q}$. For the rest of the paper, the approach is exemplified with logistic regression as $p_{\boldsymbol{\theta}}$ but it can be applied to any other predictive model, as will be recalled in the concluding section.

**Quantization as a model selection problem** As dicussed in the previous section, and emphasized in the literature review, quantization is often a preprocessing step; however, quantization can be embedded directly in the predictive model. Continuouing our logistic example, a standard information criteria such as the BIC (Schwarz (1978)) can be used to select the best quantization $\boldsymbol{q}$:

$$\hat{\boldsymbol{q}} = \underset{\boldsymbol{q} \in \mathcal{Q}}{\arg\max}\left\{\ell_{\boldsymbol{q}}(\hat{\boldsymbol{\theta}}_{\boldsymbol{q}}; (\mathbf{x}, \mathbf{y})) - \frac{1}{2}\nu_{\boldsymbol{q}}\ln(n)\right\} \tag{5}$$

where $\nu_{\boldsymbol{q}}$ is the number of continuous parameters to be estimated in the $\boldsymbol{\theta}$-parameter space. Note however that an exhaustive search of $\hat{\boldsymbol{q}} \in \mathcal{Q}$ is an intractable task due to its highly combinatorial nature. For example, with $d = 10$ categorical features with $l_j = 4$ levels each, $|\mathcal{Q}|$ is given by the Stirling number of the second kind to the power $d$, which is approx. $6 \cdot 10^{11}$. Anyway, the optimization (5) requires a new specific strategy that we describe in the next section.

**Remark on model identifiability** The shifting of cutpoints (2) anywhere strictly between two successive raw values of a given continuous feature induce the same quantization. Thus, the identifiability of such quantizations is obtained from the dataset **x** by fixing arbitrary cutpoints between successive data values, feature by feature. The continuous part of $\mathcal{Q}$ then becomes a discrete set.

## 3 THE PROPOSED NEURAL NETWORK BASED QUANTIZATION

### 3.1 A RELAXATION OF THE OPTIMIZATION PROBLEM

In this section, we propose to relax the constraints on $\boldsymbol{q}_j$ to simplify the search of $\hat{\boldsymbol{q}}$. Indeed, the derivatives of $\boldsymbol{q}_j$ are zero almost everywhere and consequently a gradient descent cannot be directly applied to find an optimal quantization.

**Smooth approximation of the quantization mapping** A classical approach is to replace the binary functions $q_{j,h}$ (see Equation (1)) by smooth parametric ones with a simplex condition, namely with $\boldsymbol{\alpha}_j = (\boldsymbol{\alpha}_{j,1}, \ldots, \boldsymbol{\alpha}_{j,m_j})$:

$$\boldsymbol{q}_{\boldsymbol{\alpha}_j}(\cdot) = \left(q_{\boldsymbol{\alpha}_{j,h}}(\cdot)\right)_{h=1}^{m_j} \text{ with } \sum_{h=1}^{m_j} q_{\boldsymbol{\alpha}_{j,h}}(\cdot) = 1 \text{ and } 0 \leq q_{\boldsymbol{\alpha}_{j,h}}(\cdot) \leq 1,$$

where functions $q_{\boldsymbol{\alpha}_{j,h}}(\cdot)$, properly defined hereafter for both continuous and categorical features, represent a fuzzy quantization in that, here, each level $h$ is weighted by $q_{\boldsymbol{\alpha}_{j,h}}(\cdot)$ instead of being selected once and for all as in (1). The resulting fuzzy quantization for all components depends on the global parameter $\boldsymbol{\alpha} = (\boldsymbol{\alpha}_1, \ldots, \boldsymbol{\alpha}_d)$ and is denoted by $\boldsymbol{q}_{\boldsymbol{\alpha}}(\cdot) = \left(\boldsymbol{q}_{\boldsymbol{\alpha}_j}(\cdot)\right)_{j=1}^{d}$. This approximation is justified by the following arguments. From a deterministic point of view, denoting by $\tilde{\mathcal{Q}}$ the space of $\boldsymbol{q}_{\boldsymbol{\alpha}}$, we have $\mathcal{Q} \subset \tilde{\mathcal{Q}}$. From a statistical point of view, under standard regularity conditions and with a suitable estimation procedure (see later for the proposed estimation procedure), we have consistency of $(\boldsymbol{q}_{\hat{\boldsymbol{\alpha}}}, \hat{\boldsymbol{\theta}})$ towards $(\boldsymbol{q}, \boldsymbol{\theta})$. From an empirical point of view, we will see in Section 4 and in particular in Figure 2, that this smooth approximation $\boldsymbol{q}_{\boldsymbol{\alpha}}$ converges towards "hard" quantizations[*] $\boldsymbol{q}$.

**For continuous features**, we set for $\boldsymbol{\alpha}_{j,h} = (\alpha_{j,h}^0, \alpha_{j,h}^1) \in \mathbb{R}^2$

$$q_{\boldsymbol{\alpha}_{j,h}}(\cdot) = \frac{\exp(\alpha_{j,h}^0 + \alpha_{j,h}^1 \cdot)}{\sum_{g=1}^{m_j} \exp(\alpha_{j,g}^0 + \alpha_{j,g}^1 \cdot)}$$

where $\boldsymbol{\alpha}_{j,m_j}$ is set to $(0,0)$ for identifiability reasons.

**For categorical features**, we set for $\boldsymbol{\alpha}_{j,h} = (\alpha_{j,h}(1), \ldots, \alpha_{j,h}(l_j)) \in \mathbb{R}^{l_j}$

$$q_{\boldsymbol{\alpha}_{j,h}}(\cdot) = \frac{\exp\left(\alpha_{j,h}(\cdot)\right)}{\sum_{g=1}^{m_j} \exp\left(\alpha_{j,g}(\cdot)\right)}$$

where $l_j$ is the number of levels of the categorical feature $x_j$.

**Parameter estimation** With this new fuzzy quantization, the logistic regression for the predictive task is then expressed as

$$\ln\left(\frac{p_{\boldsymbol{\theta}}(1|\boldsymbol{q}_{\boldsymbol{\alpha}}(\boldsymbol{x}))}{1 - p_{\boldsymbol{\theta}}(1|\boldsymbol{q}_{\boldsymbol{\alpha}}(\boldsymbol{x}))}\right) = \theta_0 + \sum_{j=1}^{d} \boldsymbol{\theta}_j' \cdot \boldsymbol{q}_{\boldsymbol{\alpha}_j}(x_j), \tag{6}$$

where $\boldsymbol{q}$ has been replaced by $\boldsymbol{q}_{\boldsymbol{\alpha}}$ from Equation (3). Note that as $\boldsymbol{q}_{\boldsymbol{\alpha}}$ is a sound approximation of $\boldsymbol{q}$ (see above), this logistic regression in $\boldsymbol{q}_{\boldsymbol{\alpha}}$ is consequently a good approximation of the logistic regression in $\boldsymbol{q}$ from Equation (3). The relevant log-likelihood is here

$$\ell_{\boldsymbol{q}_{\boldsymbol{\alpha}}}(\boldsymbol{\theta}; (\mathbf{x}, \mathbf{y})) = \sum_{i=1}^{n} \ln p_{\boldsymbol{\theta}}(y_i | \boldsymbol{q}_{\boldsymbol{\alpha}}(\boldsymbol{x}_i)) \tag{7}$$

---

[*]Up to a permutation on the labels $h = 1 \ldots m_j$ to recover the ordering in $C_{j,h}$ (see Eq. (2)).

and can be used as a tractable substitute for (4) to solve the original optimization problem (5), where now both $\boldsymbol{\alpha}$ and $\boldsymbol{\theta}$ have to be estimated, which is discussed in the next section. We wish to maximize the log-likelihood (6) which would yield parameters $(\hat{\boldsymbol{\alpha}}, \hat{\boldsymbol{\theta}})$; these are consistent if the model is well-specified (*i.e.* there is a "true" quantization under classical regularity conditions). To "push" $\widetilde{\mathcal{Q}}$ further into $\mathcal{Q}$, we deduce $\hat{\boldsymbol{q}}$ from a *maximum a posteriori* procedure applied to $\boldsymbol{q}_{\hat{\boldsymbol{\alpha}}}$:

$$\hat{q}_{j,h}(x_j) = 1 \text{ if } h = \underset{1 \leq h' \leq m_j}{\arg \max}\, q_{\hat{\boldsymbol{\alpha}}_{j,h'}}, 0 \text{ otherwise.} \tag{8}$$

If there are several levels $h$ that satisfy (8), we simply take the level that corresponds to smaller values of $x_j$ to be in accordance with the definition of $C_{j,h}$ in Equation (2). This *maximum a posteriori* principle will be exemplified in Figure 2 on simulated data.

## 3.2 A NEURAL NETWORK-BASED ESTIMATION STRATEGY

**Neural network architecture**  To estimate parameters $\boldsymbol{\alpha}$ and $\boldsymbol{\theta}$ in the model (6), a particular neural network architecture can be used. The most obvious part is the output layer that must produce $p_{\boldsymbol{\theta}}(1|\boldsymbol{q}_{\boldsymbol{\alpha}}(\boldsymbol{x}))$ which is equivalent to a densely connected layer with a sigmoid activation $\sigma(\cdot)$.

For a continuous feature $x_j$ of $\boldsymbol{x}$, the combined use of $m_j$ neurons including affine transformations and softmax activation obviously yields $\boldsymbol{q}_{\boldsymbol{\alpha}_j}(x_j)$. Similarly, an input categorical feature $x_j$ with $l_j$ levels is equivalent to $l_j$ binary input neurons (presence or absence of the factor level). These $l_j$ neurons are densely connected to $m_j$ neurons without any bias term and a softmax activation. The softmax outputs are next aggregated via the summation in model (6), say $\Sigma_{\boldsymbol{\theta}}$ for short, and then the sigmoid function $\sigma$ gives the final output. All in all, the proposed model is straightforward to optimize with a simple neural network, as shown in Figure 1.

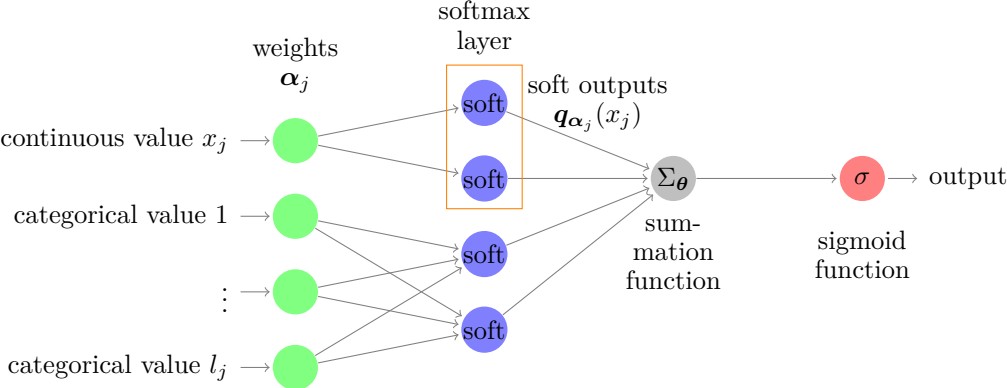

Figure 1: Proposed shallow architecture to maximize (7).

**Stochastic gradient descent as a quantization provider**  By relying on a stochastic gradient descent, the smoothed likelihood (7) can be maximized over $(\boldsymbol{\alpha}, \boldsymbol{\theta})$. The results should be close to the maximizers of the original likelihood (4) if the model is well-specified, when there is a true underlying quantization. In the mis-specified model case, there is no such guarantee. Therefore, to be more conservative, we evaluate at each training epoch $(t)$ the quantization $\hat{\boldsymbol{q}}^{(t)}$ resulting from the *maximum a posteriori* procedure explicited in Equation (8), then classicaly estimate the logistic regression parameter *via* maximum likelihood, as done in Equation (4):

$$\boldsymbol{\theta}^{(t)} = \underset{\boldsymbol{\theta}}{\arg \min}\, \ell_{\boldsymbol{q}^{(t)}}(\boldsymbol{\theta}; (\mathbf{x}, \mathbf{y}))$$

and the resulting $\text{BIC}^{(t)}$ as in (5). If $T$ is a given maximum number of iterations of the stochastic gradient descent algorithm, the quantization retained at the end is then deter-

mined by the optimal epoch

$$t_* = \arg\min_{t \in \{1,\dots,T\}} \text{BIC}^{(t)}.$$

**Choosing an appropriate number of levels** Concerning now the number of intervals or factor levels $\boldsymbol{m} = (m_j)_1^d$, they have also to be estimated since in practice they are unknown. Looping over all candidates $\boldsymbol{m}$ is intractable. But in practice, by relying on the *maximum a posteriori* procedure developed in Equation (8), we might drop a lot of unseen factor levels, *e.g.* if $q_{\boldsymbol{\alpha}_{j,h}}(x_{i,j}) \ll 1$ for all training observations $x_{i,j}$, the level $h$ "vanishes", *i.e.* $\hat{q}_{j,h} = 0$. In practice, we recommend to start with a user-chosen $\boldsymbol{m} = \boldsymbol{m}_{\max}$ and we will see in the experiments of Section 4 that the proposed approach is able to explore small values of $\boldsymbol{m}$ and to select a value $\hat{\boldsymbol{m}}$ drastically smaller than $\boldsymbol{m}_{\max}$. This phenomenon, which reduces the computational burden of the quantization task, is also illustrated in the next section.

## 4 NUMERICAL EXPERIMENTS

This section is divided into three complementary parts to assess the validity of our proposal, that we call hereafter *glmdisc*. First, simulated data are used to evaluate its ability to recover the true data generating mechanism. Second, the predictive quality of the new learned representation approach is illustrated on several classical benchmark datasets from the UCI library. Third, we use it on *Credit Scoring* datasets provided by Credit Agricole Consumer Finance, a major European company in the consumer credit market. The Python notebooks of all experiments, excluding the confidential real data, can be found on the first author's website.

### 4.1 SIMULATED DATA: EMPIRICAL CONSISTENCY AND ROBUSTNESS

We focus here on discretization of continuous features (similar experiments could be conducted on categorical ones). Two continuous features $x_1$ and $x_2$ are sampled from the uniform distribution on $[0, 1]$ and discretized by using

$$\boldsymbol{q}_1(\cdot) = \boldsymbol{q}_2(\cdot) = (\mathbb{1}_{]-\infty,1/3]}(\cdot), \mathbb{1}_{]1/3,2/3]}(\cdot), \mathbb{1}_{]2/3,\infty]}(\cdot)).$$

Here, following (2), we have $d = 2$ and $m_1 = m_2 = 3$ and the cutpoints are $c_{j,1} = 1/3$ and $c_{j,2} = 2/3$ for $j = 1, 2$. Setting $\boldsymbol{\theta} = (0, -2, 2, 0, -2, 2, 0)$, the target feature $y$ is then sampled from $p_{\boldsymbol{\theta}}(\cdot|\boldsymbol{q}(\boldsymbol{x}))$ via the logistic model (3).

From the *glmdisc* algorithm, we studied three cases:

(a) First, the quality of the cutoff estimator $\hat{c}_{j,2}$ of $c_{j,2} = 2/3$ is assessed when the starting maximum number of intervals per discretized continuous feature is set to its true value $m_1 = m_2 = 3$;

(b) Second, we estimated the number of intervals $\hat{m}_1$ of $m_1 = 3$ when the starting maximum number of intervals per discretized continuous feature is set to $m_{\max} = 10$;

(c) Last, we added a third feature $x_3$ also drawn uniformly on $[0, 1]$ but uncorrelated to $y$ and estimated the number $\hat{m}_3$ of discretization intervals selected for $x_3$. The reason is that a non-predictive feature which is discretized or grouped into a single value is *de facto* excluded from the model, and this is a positive side effect.

From a statistical point of view, experiment (a) assesses the empirical consistency of the estimation of $C_{j,h}$, whereas experiments (b) and (c) focus on the consistency of the estimation of $m_j$. The results are summarized in Table 1 where 95% confidence intervals (CI) are given, with a varying sample size. Note in particular that the slight underestimation in (b) is a classical consequence of the BIC criterion on small samples.

### 4.2 BENCHMARK DATA

To test further the effectiveness of *glmdisc* in a predictive setting, we gathered 6 datasets from the UCI library: the Adult dataset ($n = 48,842$, $d = 14$), the Australian dataset

| Sample size | (a) $\hat{c}_{j,2}$ | (b) $\hat{m}_1$ | (c) $\hat{m}_3$ |
|---|---|---|---|
| $n = 1,000$ | $[0.656, 0.666]$ | $[2.679, 2.941]$ | $[1.326, 1.554]$ |
| $n = 10,000$ | $[0.666, 0.666]$ | $[3.000, 3.000]$ | $[1.399, 1.621]$ |

Table 1: (a) CI of $\hat{c}_{j,2}$ for $c_{j,2} = 2/3$. (b) CI of $\hat{m}$ for $m_1 = 3$. (c) CI of $\hat{m}_3$ for $m_3 = 1$.

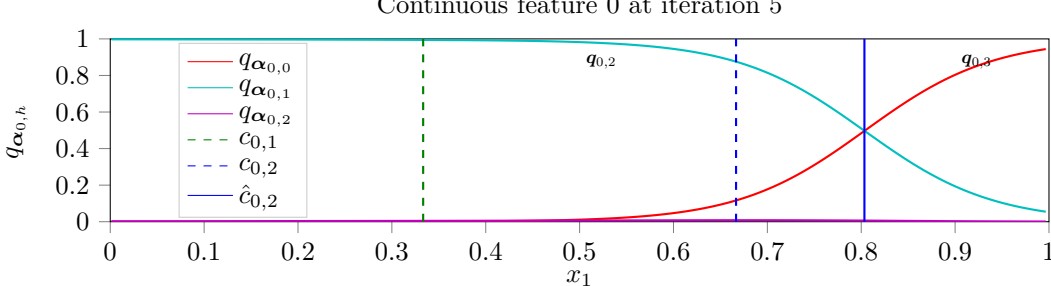

(a) Quantization $\hat{q}_1^{(t)}(x_1)$ resulting from the thresholding (8) at iterations $t = 5$ and $m_{\text{start}} = 3$.

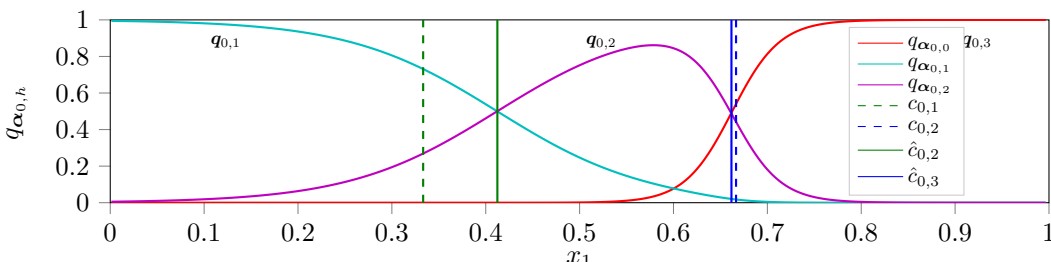

(b) Quantizations $\hat{q}_1^{(t)}(x_1)$ resulting from the thresholding (8) at iterations $t = 300$ and $m_{\text{start}} = 3$.

Figure 2: Quantizations $\hat{q}_1^{(t)}(x_1)$ of experiment (a) resulting from the thresholding (8).

($n = 690$, $d = 14$), the Bands dataset ($n = 512$, $d = 39$), the Credit-screening dataset ($n = 690$, $d = 15$), the German dataset ($n = 1,000$, $d = 20$) and the Heart dataset ($n = 270$, $d = 13$). Each of these datasets have mixed (continuous and categorical) features and a binary response to predict. To get more information about these datasets, their respective features, and the predictive task associated with them, readers may refer to the UCI website[†].

Now that we made sure that our approach is empirically consistent, *i.e.* it is able to find the true quantization in a well-specified setting, we wish to verify our claim that embedding the learning of a good quantization in the predictive task *via glmdisc* is better than other methods that rely on *ad hoc* criteria. As we were primarily interested in logistic regression, we will compare our approach to a naïve linear logistic regression, a logistic regression on continuous discretized data using the now standard MDLP algorithm from Fayyad & Irani (1993) and categorical grouped data using $\chi^2$ tests of independence between each pair of factor levels and the target in the same fashion as the ChiMerge discretization algorithm proposed by Kerber (1992). As the original use case stems from *Credit Scoring*, we use the performance metric usually monitored by *Credit Scoring* practitioners, which is the Gini coefficient, directly related to the Area Under the ROC Curve (Gini $= 2 \times \text{AUC} - 1$).

Table 2 shows our approach yields significantly better results on these rather small datasets where the added flexibility of quantization might help the predictive task.

---

[†]Dheeru & Karra Taniskidou (2017) : http://archive.ics.uci.edu/ml

| Dataset | Additive linear logistic regression | *ad hoc* methods | Our proposal: *glmdisc* |
|---|---|---|---|
| Adult | 81.5 | **84.8** | 81.0 |
| Australian | 73.6 | 65.9 | **92.1** |
| Bands | 48.1 | 45.4 | **58.5** |
| Credit-screening | 81.3 | 88.5 | **93.4** |
| German | 52.1 | 57.9 | **70.4** |
| Heart | 79.4 | 76.0 | **84.0** |

Table 2: Gini indices of our proposed representation learning algorithm *glmdisc* and two baselines: a "naïve" logistic regression and *ad hoc* methods (MDLP / $\chi^2$ tests) obtained on several benchmark datasets from the UCI library.

| Portfolio | Additive linear logistic regression | Current performance | *ad hoc* methods | Our proposal: *glmdisc* |
|---|---|---|---|---|
| Automobile loans | 58.8 | 55.6 | **60.4** | 55.0 |
| Renovation loans | 52.3 | 50.9 | **54.2** | 49.6 |
| Standard loans | 39.7 | 37.1 | **46.3** | 41.0 |
| Revolving loans | 61.5 | 58.5 | **62.9** | 60.3 |
| Mass retail loans | 52.5 | 48.7 | **63.7** | 61.3 |
| Electronics loans | 52.6 | 55.8 | 61.6 | **67.0** |

Table 3: Gini indices of our proposed representation learning algorithm *glmdisc*, the two baselines of Table 2 and the current scorecard (manual / expert representation) obtained on several portfolios of Credit Agricole Consumer Finance.

### 4.3 *Credit Scoring* DATA

Discretization, grouping and interaction screening are preprocessing steps relatively "manually" performed in the field of *Credit Scoring*, using $\chi^2$ tests for each feature or so-called Weights of Evidence (Zeng (2014)). This back and forth process takes a lot of time and effort and provides no particular statistical guarantee.

Table 3 shows Gini coefficients of several portfolios for which there are $n = 50,000$, $n = 30,000$, $n = 50,000$, $n = 100,000$, $n = 235,000$ and $n = 7,500$ clients respectively and $d = 25$, $d = 16$, $d = 15$, $d = 14$, $d = 14$ and $d = 16$ features respectively. Approximately half of these features were categorical, with a number of factor levels ranging from 2 to 100.

We compare the rather manual, in-house approach that yields the current performance, the naïve linear logistic regression and *ad hoc* methods introduced in the previous section and finally our *glmdisc* proposal. Beside the classification performance, interpretability is maintained and unsurprisingly, the learned representation comes often close to the "manual" approach: for example, the complicated in-house coding of job types is roughly grouped by *glmdisc* into *e.g.* "worker", "technician", *etc.* Notice that even if the "naïve" logistic regression reaches some very decent predictive results, its poor interpretability skill (no quantization at all) excludes it from standard use in the company.

The usefulness of discretization and grouping is clear on *Credit Scoring* data and although *glmdisc* does not always perform significantly better than the manual approach, it allows practitioners to focus on other tasks by saving a lot of time, as was already stressed out. As a rule of thumb, a month is generally allocated to data pre-processing for a single data scientist working on a single scorecard. On Google Collaboratory, and relying on Keras (Chollet et al. (2015)) and Tensorflow (Abadi et al. (2015)) as a backend, it took less than an hour to perform discretization and grouping for all datasets.

## 5 CONCLUDING REMARKS

Feature quantization (discretization for continuous features, grouping of factor levels for categorical ones) in a supervised multivariate classification is a recurring problem in many

industrial contexts. This setting was formalized as a highly combinatorial representation learning problem and a new algorithmic approach, named *glmdisc*, has been proposed as a sensible approximation of a classical statistical information criterion.

This algorithm relies on the use of a softmax approximation of each discretized or grouped feature. This proposal can alternatively be replaced by any other univariate multiclass predictive model, which makes it flexible and adaptable to other problems. Prediction of the target feature, given quantized features, was exemplified with logistic regression, although here as well, it can be swapped with any other supervised classification model. An estimation strategy putting neural networks' good computational properties to use was introduced while maintaining the interpretability necessary to some fields of application.

The experiments showed that, as was sensed empirically by statisticians in the field of *Credit Scoring*, discretization and grouping can indeed provide better models than standard logistic regression. This novel approach allows practitioners to have a fully automated and statistically well-grounded tool that achieves better performance than *ad hoc* industrial practices at the price of decent computing time but much less of the practitioner's valuable time.

As described in the introduction, logistic regression is additive in its inputs which does not allow to take into account conditional dependency, as stated by Berry et al. (2010). This problem is often dealt with by sparsely introducing "interactions", *i.e.* products of two features. This leads again to a model selection challenge on a highly combinatorial discrete space that could be solved with a similar approach. In a broader context with no restriction on the predictive model, Tsang et al. (2018) already made use of neural networks to estimate the presence or absence of statistical interactions. The parsimonious addition of pairwise interactions among quantized features, that might influence the quantization process introduced in this work, is a future area of research.

### ACKNOWLEDGMENTS

The authors are thankful to Credit Agricole Consumer Finance for providing data and funding through a CIFRE PhD, made possible by the Association Nationale de la Recherche et la Technologie (ANRT). Many thanks also go to Pascal Germain for his useful insights on the architecture and optimization procedures of neural networks.

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

# ICLR 2019 - Additional material to 'Feature quantization for parsimonious and interpretable predictive models'

November 23, 2018

## 1 Continuous and categorical data generation

```
In [1]: import random
        import numpy as np
        import pandas as pd
        import sklearn as sk
        import sklearn.preprocessing
        import matplotlib.pyplot as plt
        plt.rc('text', usetex=True)
        plt.rc('font', **{'family': "sans-serif"})
        params = {
            'text.latex.preamble':
            [r'\usepackage{amsmath}', r'\usepackage{amsfonts}', r'\usepackage{bm}']
        }
        plt.rcParams.update(params)
        from matplotlib2tikz import save as tikz_save

In [2]: from IPython.display import set_matplotlib_formats
        set_matplotlib_formats('pdf', 'png')
        plt.rcParams['savefig.dpi'] = 75

        plt.rcParams['figure.autolayout'] = False
        plt.rcParams['figure.figsize'] = 10, 6
        plt.rcParams['axes.labelsize'] = 18
        plt.rcParams['axes.titlesize'] = 20
        plt.rcParams['font.size'] = 16
        plt.rcParams['lines.linewidth'] = 2.0
        plt.rcParams['lines.markersize'] = 8
        plt.rcParams['legend.fontsize'] = 14

        plt.rcParams['text.usetex'] = True
        plt.rcParams['font.family'] = "serif"
        plt.rcParams['font.serif'] = "cm"

In [3]: # n: sample size; d1: continuous features' dimension;
        # x_quant: U([0,1]); xd_quant: "true" generating data of Y
```

```
n = 1000
d1 = 2
cuts = ([0, 0.333, 0.666, 1])
```

In [4]:
```python
# x_quant is cut in thirds into xd_quant
x_quant = np.array(np.random.uniform(size=(n, d1)))
xd_quant = np.ndarray.copy(x_quant)
for i in range(d1):
    xd_quant[:, i] = pd.cut(x_quant[:, i], bins=cuts, labels=[0, 1, 2])
```

## 2   Target feature generation following a true discrete logistic regression

In [5]:
```python
# The true logistic regression parameter is fixed
if d1 > 0:
    theta_quant = np.array([[0] * d1] * (len(cuts) - 1))
    theta_quant[1, :] = 2
    theta_quant[2, :] = -2
```

In [6]:
```python
# The log odd probabilities of Y given each x_i can be exactly calculated
log_odd = np.array([0] * n)
for i in range(n):
    for j in range(d1):
        log_odd[i] += theta_quant[int(xd_quant[i, j]), j]
```

In [7]:
```python
# Y is then drawn from this pdf
p = 1 / (1 + np.exp(-log_odd))
y = np.random.binomial(1, p)
```

## 3   Some simple visualization

In [8]:
```python
# The true features generating Y are discrete so the number
# of points per unique value of log-odd ratio must be calculated
import collections
points_bokeh_size = np.zeros((len(np.unique(p)), 2))
points_bokeh_y = np.zeros((len(np.unique(p)), 2))
points_bokeh_y[:, 1] = 1
for i in range(len(np.unique(p))):
    points_bokeh_size[i, 0] = 100 * collections.Counter(
        y[np.where(p == np.unique(p)[i])])[0] / n
    points_bokeh_size[i, 1] = 100 * collections.Counter(
        y[np.where(p == np.unique(p)[i])])[1] / n
```

In [9]:
```python
# The red curve is the classical sigmoid curve; green points are of class 1
# and their relative size represent how many samples take these values;
# likewise, blue points are of class 0.
plt.scatter(
    np.unique(log_odd),
```

```
        points_bokeh_y[:, 0],
        marker='o',
        s=points_bokeh_size[:, 0],
        c="blue")
    plt.scatter(
        np.unique(log_odd),
        points_bokeh_y[:, 1],
        marker='o',
        s=points_bokeh_size[:, 1],
        c="green")
    plt.plot(np.unique(log_odd), np.unique(p), color="red")
    plt.show()
```

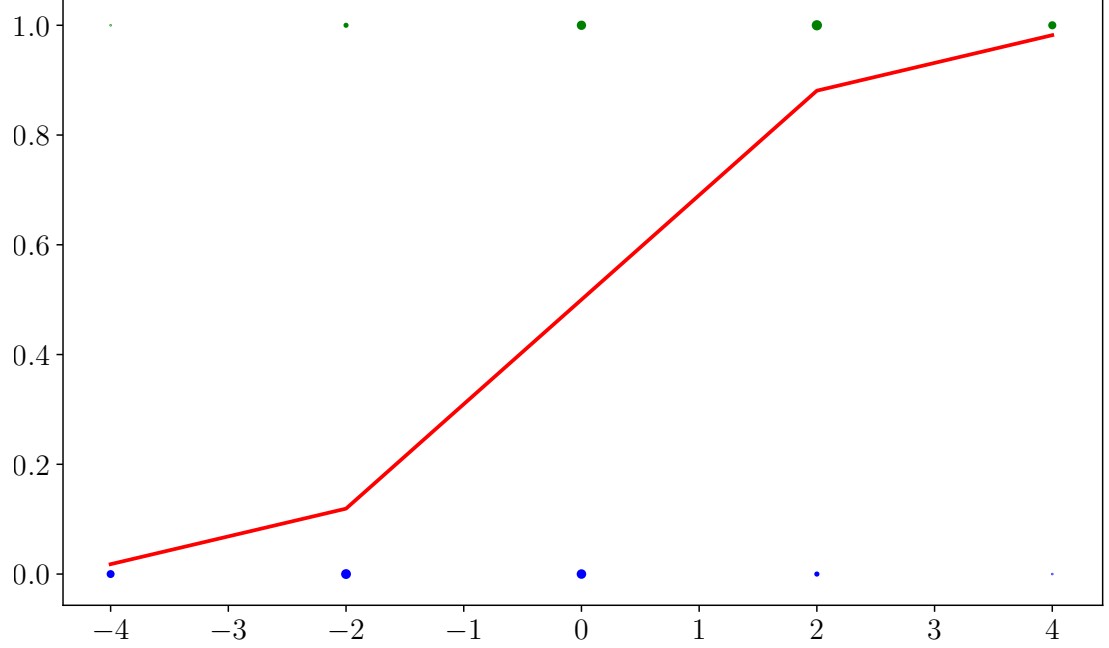

```
In [10]: # The original value of the continuous features and their generated Y
         # can be plotted; the provided cuts (.333 and .666) are clearly visible
         # and separate different proportions of classes 0 and 1.
         # Clearly, such complicated boundaries could not be determined via
         # a classical linear logistic regression.
         plt.scatter(
             x_quant[0:10000, 0],
             x_quant[0:10000, 1],
             marker='o',
             s=points_bokeh_size[:, 0],
             c=(np.where(y[0:10000] == 0, "green", "red")))
         plt.hlines(0.333, 0, 1, colors='blue')
```

```
plt.hlines(0.666, 0, 1, colors='blue')
plt.vlines(0.333, 0, 1, colors='blue')
plt.vlines(0.666, 0, 1, colors='blue')
plt.show()
```

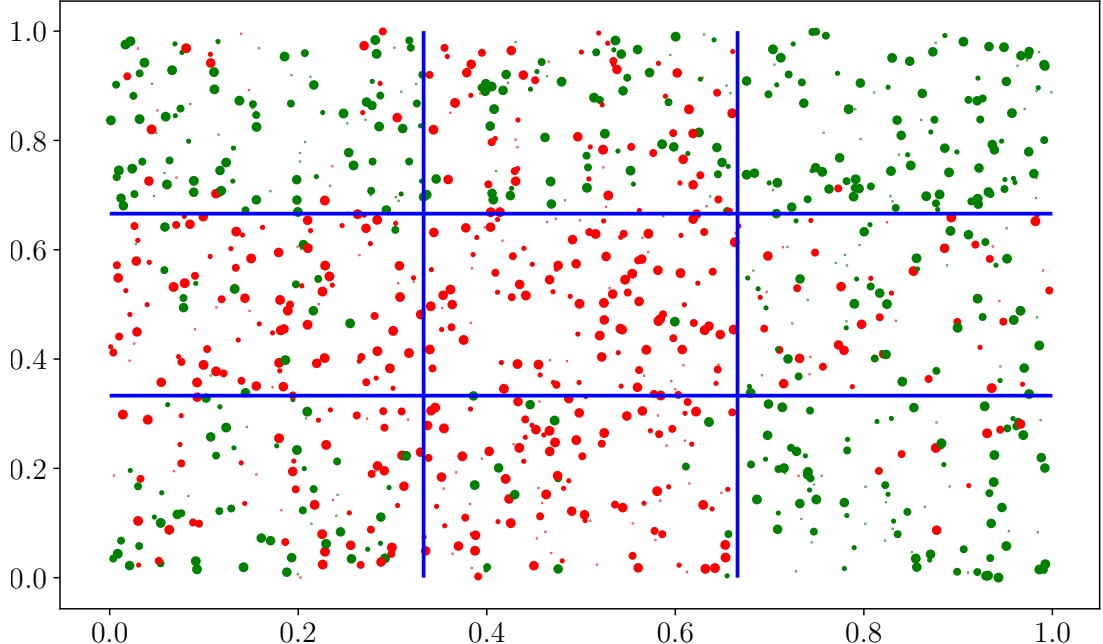

We could do a similar graphic for categorical features.

# 4   Some Math

We generated some data $= (x_{i,j})_{1 \le i \le n, 1 \le j \le d1+d2}$.

We preprocessed this data with a specified $q^\star = (q_j)_1^{d1}$ s.t.:

$1 \le j \le d1, q_j^\star(x_j) = (\mathbb{1}_{]-\infty;0.333]}(x_j), \mathbb{1}_{]0.333;0.666]}(x_j), \mathbb{1}_{]0.666,\infty[}(x_j))$

We then went on to prespecify a regression coefficient $\theta^\star$ and generated the target feature $y$ s.t.:

$Y \sim p_{\theta^\star}(\cdot | f^\star(x))$ where $\ln \left( \dfrac{p(1|q(x);\theta)}{1 - p(1|q(x);\theta)} \right) = \theta_0 + \sum_{j=1}^{d1} \theta_j' \times q_j(x_j)$

This gives us $n$ responses $= (y_i)_1^n$.

In practice, we are just given $(,)$ and wish to recover the "true" model $q^\star, \theta^\star$. To do so in a finite sample setting, we give ourselves a model selection criterion like BIC, AIC, test set AUC, ...

All discretization / grouping schemes $q \in \mathcal{Q}$ cannot be tested (very high combinatorics) so that there is a need for additional assumptions / modelling + estimation procedure to reduce the computational burden and have at least some statistical guarantees (in $n$) to recover $q^\star$.

N.B.: If the true data generating mechanism is not a discrete logistic regression, we get a representation of $X$ that yields the "best" logistic regression (on discrete data w.r.t. a given criterion / loss), which is in practice very useful in the field of *Credit Scoring* (initial use case of this work).

We rely on a softmax approximation of $q$, i.e. $q_{\alpha_{j,h}} \propto \exp(\alpha_{j,h,0} + \alpha_{j,h,1} x_j)$, where the likelihood of $(q, \theta)$ is "swapped" for $(\alpha, \theta)$.

**Refer to the paper for the proper justifications.**

## 5    Neural network architecture

To estimate $\theta$ and $\alpha$, we can use shallow neural networks by stating that: - If we allow $q_j$ to take $m_j$ values (we discretize in $m_j$ intervals), then we need $m_j$ softmax neurons per raw continuous feature $x_j$. Other features shall not be connected to these neurons; - If we wish to group $x_j$ (categorical) into $m_j \leq o_j$ levels, we one-hot encode each factor level and have an input neuron per level; we densely connect each of these neurons to $m_j$ softmax neurons.

**Again, refer to the paper for the proper justifications.**

```python
In [11]: # We rely on Keras for all calculations and the TensorFlow backend.
         import tensorflow as tf
         sess = tf.Session()
         from keras import backend as K
         K.set_session(sess)
```

```
/anaconda3/lib/python3.6/site-packages/h5py/__init__.py:36: FutureWarning: Conversion of the sec
  from ._conv import register_converters as _register_converters
Using TensorFlow backend.
```

```python
In [18]: # According to the previous section, each feature has its own
         # densely connected hidden layer

         from keras import *
         from keras.layers import *
         from keras.callbacks import LambdaCallback, Callback, ReduceLROnPlateau, TensorBoard
         import sklearn.linear_model

         liste_inputs_quant = [None] * d1

         liste_layers_quant = [None] * d1

         liste_layers_quant_inputs = [None] * d1

         m_quant = [5, 5]

         for i in range(d1):
             liste_inputs_quant[i] = Input((1, ))
             liste_layers_quant[i] = Dense(m_quant[i], activation='softmax')
             liste_layers_quant_inputs[i] = liste_layers_quant[i](liste_inputs_quant[i])
```

```python
In [19]: # A few utility functions are necessary to convert these parameters into
         # the pdf parametrized by alpha as described in the previous section.

         # First, we transform the hidden layer's weights into a multinomial pdf
```

```
        def from_layers_to_proba_training():

            results = [None] * (d1)

            for j in range(d1):
                results[j] = K.function([liste_layers_quant[j].input],
                                        [liste_layers_quant[j].output])(
                                            [x_quant[:, j, np.newaxis]])
            return (results)
```

 # Second, we need to affect each discretizing / grouping to the argmax of
         # the previously calculated probabilities.

```
        def evaluate_disc():
            proba = from_layers_to_proba_training()

            results = [None] * (d1)

            X_transformed = np.ones((n, 1))

            for j in range(d1):
                results[j] = np.argmax(proba[j][0], axis=1)
                X_transformed = np.concatenate(
                    (X_transformed, sk.preprocessing.OneHotEncoder().fit_transform(
                        X=results[j].reshape(-1, 1)).toarray()),
                    axis=1)

            # Last we fit a logistic regression on the discretized / grouped features
            # and evaluate its BIC

            proposed_logistic_regression = sk.linear_model.LogisticRegression(
                fit_intercept=False)

            proposed_logistic_regression.fit(X=X_transformed, y=y.reshape((n, )))

            BIC_training = 2 * sk.metrics.log_loss(
                y,
                proposed_logistic_regression.predict_proba(X=X_transformed)[:, 1],
                normalize=False
            ) + proposed_logistic_regression.coef_.shape[1] * np.log(n)

            return (BIC_training)
```

 # We concatenate all the layers (each feature has its layer; layers are
         # in two lists depending on them being quantitative or qualitative).

```python
        # We define the ouptput as a sigmoid (logistic regression).
        # We use an Adadelta optimizer.
        # We wish to optimize the log likelihood of the resulting model.

        from itertools import chain
        full_hidden = concatenate(
            list(chain.from_iterable([liste_layers_quant_inputs])))
        output = Dense(1, activation='sigmoid')(full_hidden)
        model = Model(
            inputs=list(chain.from_iterable([liste_inputs_quant])), outputs=[output])
        opt = optimizers.RMSprop(lr=0.5, rho=0.9, epsilon=None, decay=0.0)
        model.compile(loss='binary_crossentropy', optimizer=opt, metrics=['accuracy'])
```

In [22]: `# At each training epoch, we must evaluate the quality of the derived`

```python
        class LossHistory(Callback):
            def on_train_begin(self, logs={}):
                self.losses = []
                self.best_criterion = float("inf")
                self.best_outputs = []
                self.epochs = 0

            def on_epoch_end(self, batch, logs={}):
                self.losses.append(evaluate_disc())
                self.epochs += 1
                outputs = []

                for j in range(d1):
                    outputs.append(
                        K.function([liste_layers_quant[j].input],
                                    [liste_layers_quant[j].output])(
                                        [x_quant[:, j, np.newaxis]]))
                    if self.epochs in [1, 3, 5, 10, 20, 50, 100, 200]:
                        plt.xlim((0, 1))
                        plt.ylim((0, 1))
                        for k in range(outputs[j][0].shape[1]):
                            plt.plot(
                                np.sort(x_quant[:, j]),
                                outputs[j][0][np.argsort(x_quant[:, j]), k],
                                label=r'${q}_{\bm{\alpha}_{' + str(j) + ',' + str(k) +
                                '}}$',
                                color=['b', 'g', 'r', 'c', 'm', 'y', 'k', 'w'][k + 2])
                        plt.title('Continuous feature ' + str(j) + ' at iteration ' +
                                str(self.epochs))
                        plt.ylabel(r'${q}_{\bm{\alpha}_{' + str(j) + ',h}}$')
                        plt.xlabel(r'$x_1$')
                        plt.axvline(
```

```python
        x=0.33333,
        label=r'$c_{' + str(j) + ',1}$',
        color='g',
        linestyle='--')
    plt.axvline(
        x=0.66666,
        label=r'$c_{' + str(j) + ',2}$',
        color='b',
        linestyle='--')
    plt.text(
        0.10,
        0.95,
        r'$\boldsymbol{q}_{' + str(j) + ',1}$',
        fontsize=14,
        verticalalignment='top')
    plt.text(
        0.50,
        0.95,
        r'$\boldsymbol{q}_{' + str(j) + ',2}$',
        fontsize=14,
        verticalalignment='top')
    plt.text(
        0.90,
        0.95,
        r'$\boldsymbol{q}_{' + str(j) + ',3}$',
        fontsize=14,
        verticalalignment='top')
    df = pd.DataFrame(
        np.column_stack((x_quant[:, j],
                         np.argmax(outputs[j][0], axis=1))),
        columns=['continuous', 'discrete'])
    grouped_data = df.groupby(
        'discrete', as_index=False)['continuous']
    results = [grouped_data.max(), grouped_data.min()]
    results[0]['continuous'].values[::-1].sort()
    results[1]['continuous'].values[::-1].sort()
    moyennes = np.column_stack(
        (results[0]['continuous'].values[1:, np.newaxis],
         results[1]['continuous'].values[:-1, np.newaxis]
        )).mean(axis=1)
    for k in reversed(range(moyennes.shape[0])):
        plt.axvline(
            x=moyennes[k],
            label=r'$\hat{c}_{' + str(j) + ',' +
            str(moyennes.shape[0] - k + 1) + '}$',
            color=['b', 'g', 'r', 'c', 'm', 'y', 'k', 'w'][k],
            linestyle='-')
    plt.legend(loc='center right')
```

```
                plt.show()

            if self.losses[-1] < self.best_criterion:
                self.best_weights = []
                self.best_outputs = []
                self.best_criterion = self.losses[-1]
                for j in range(d1):
                    self.best_weights.append(liste_layers_quant[j].get_weights())
                    self.best_outputs.append(
                        K.function([liste_layers_quant[j].input],
                                   [liste_layers_quant[j].output])(
                                       [x_quant[:, j, np.newaxis]]))
```

```
In [23]: from matplotlib import colors as mcolors
         colors = list(mcolors.CSS4_COLORS.keys())
         history = LossHistory()

         callbacks = [
             ReduceLROnPlateau(
                 monitor='loss',
                 factor=0.5,
                 patience=10,
                 verbose=0,
                 mode='auto',
                 min_delta=0.0001,
                 cooldown=0,
                 min_lr=0), history
         ]
         model.fit(
             list(chain.from_iterable([list(x_quant.T)])),
             y,
             epochs=200,
             batch_size=30,
             verbose=0,
             callbacks=callbacks)
```

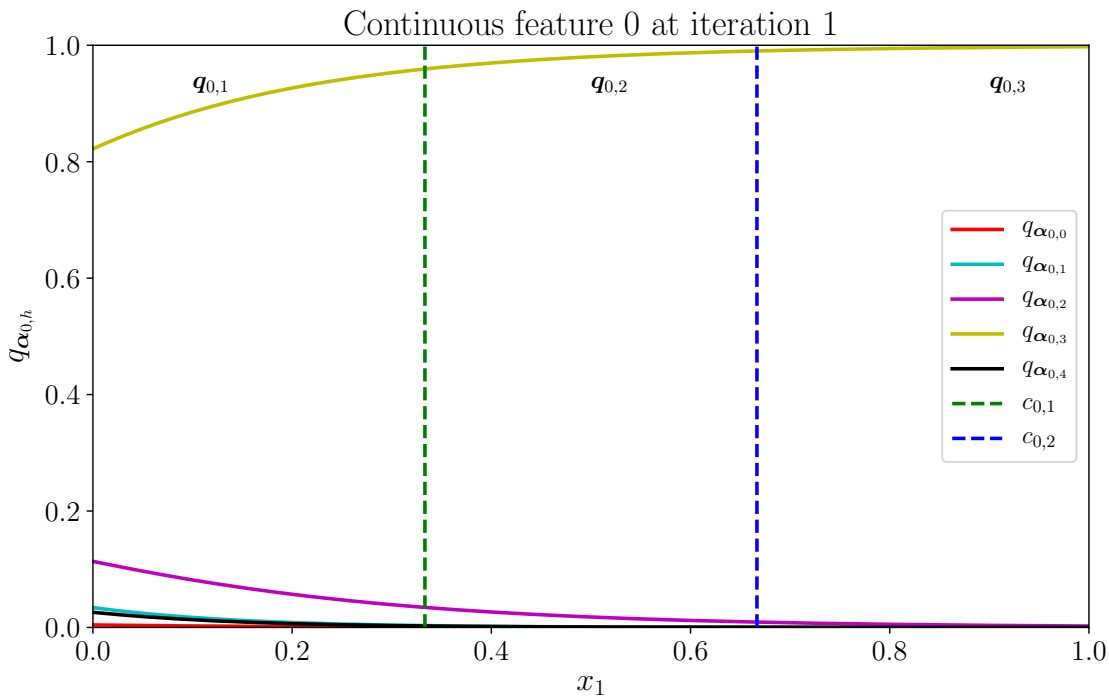

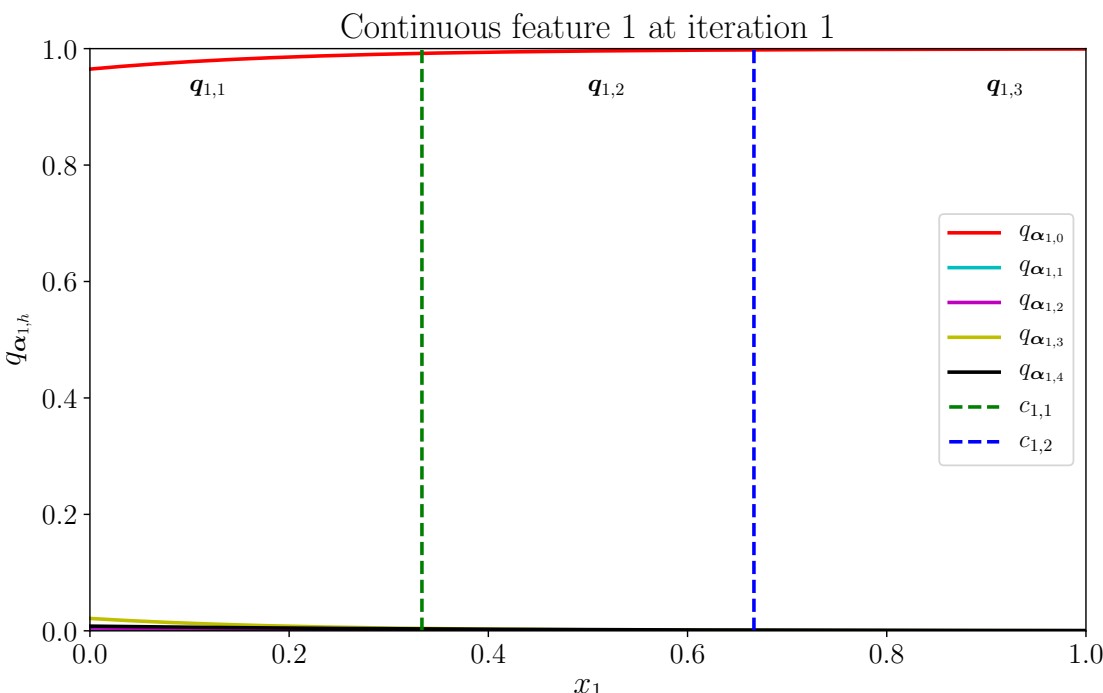

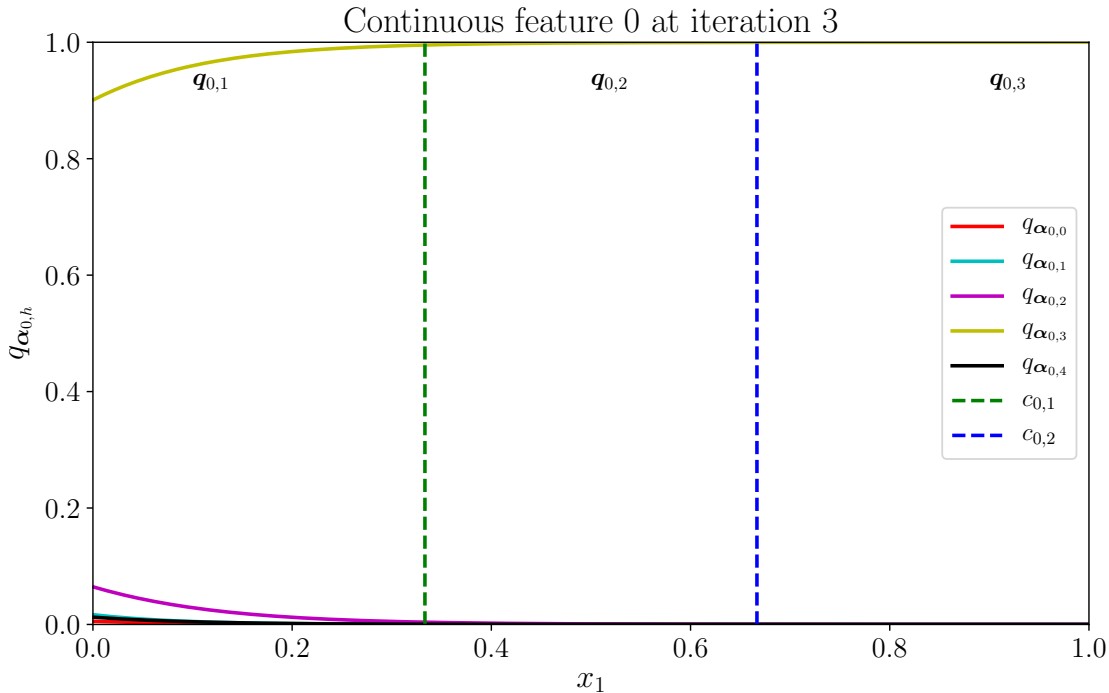

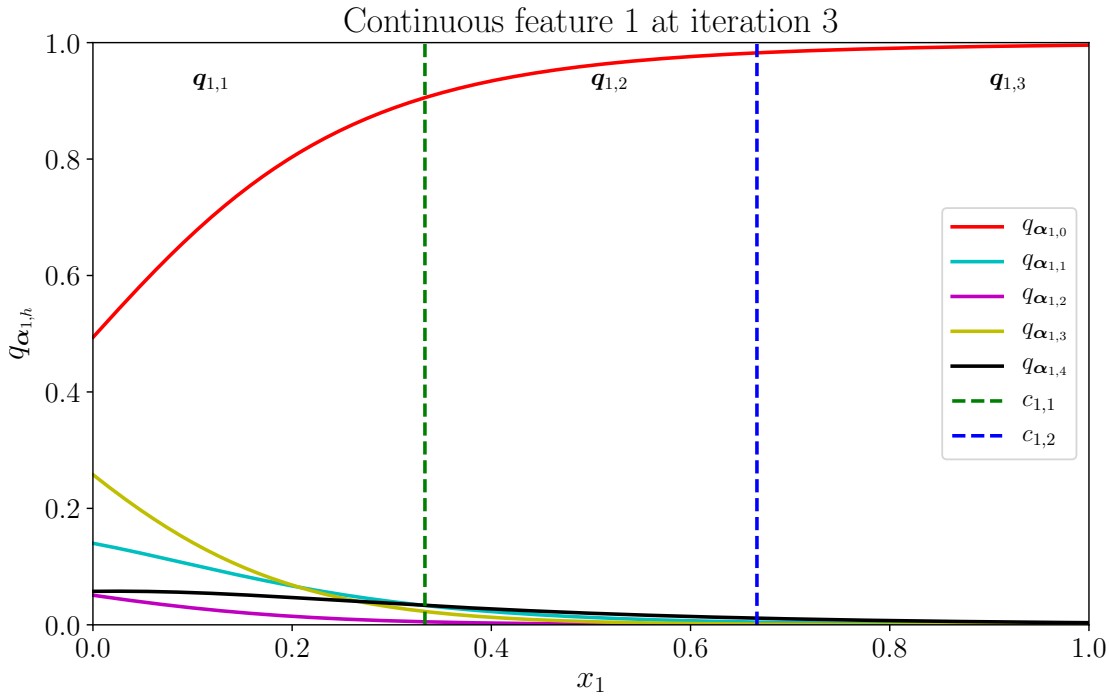

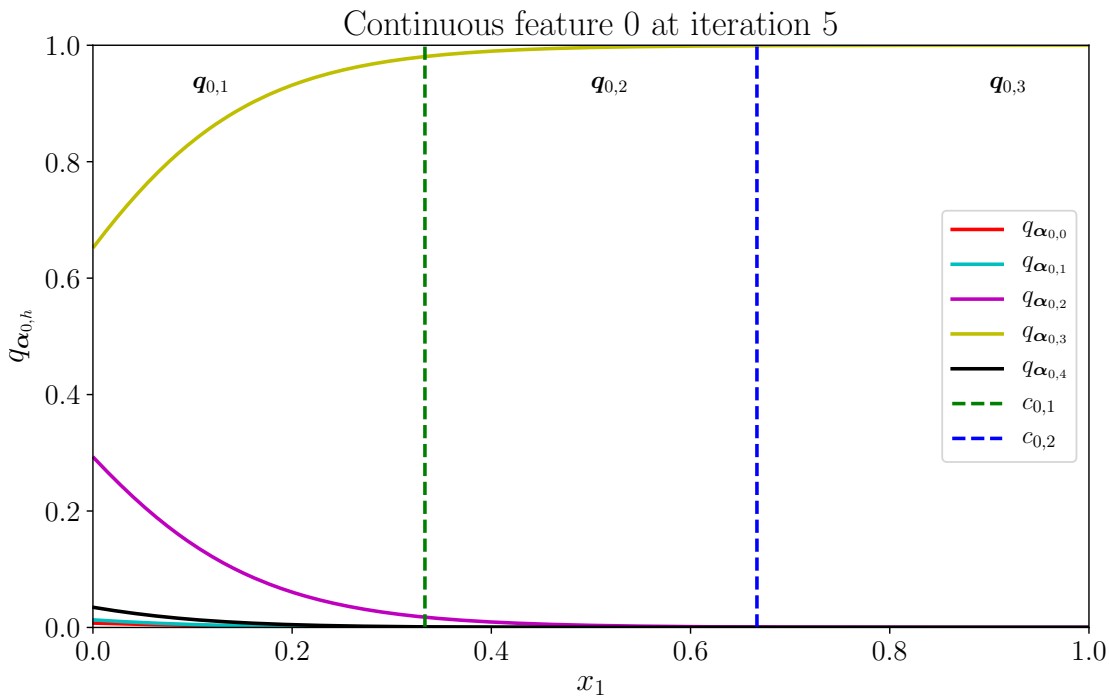

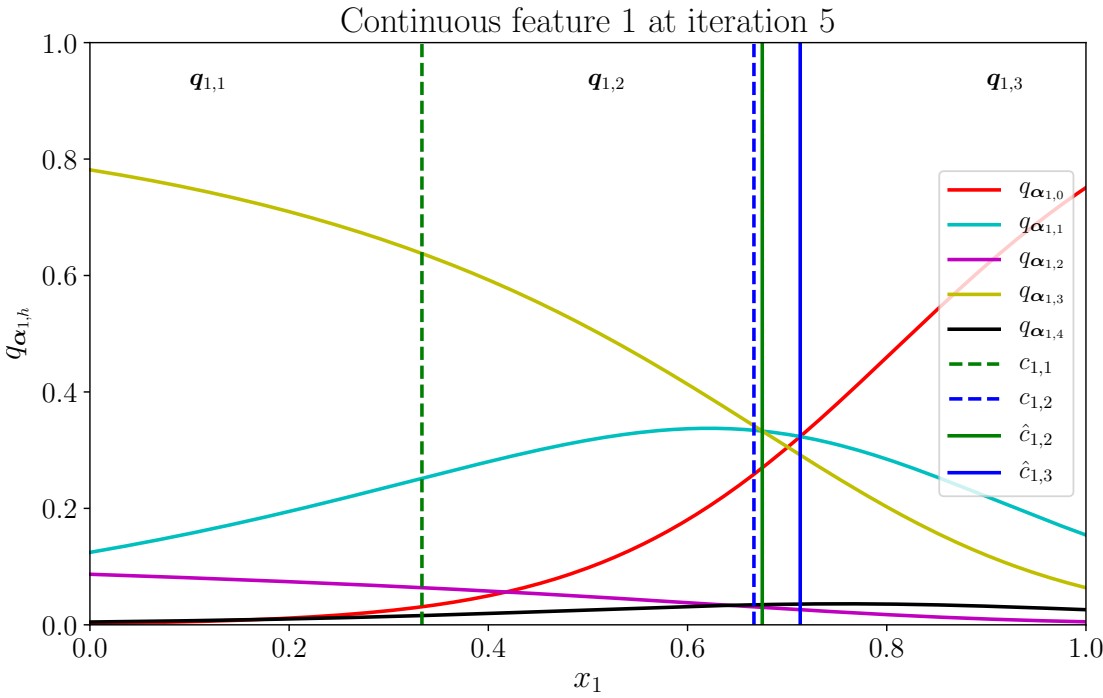

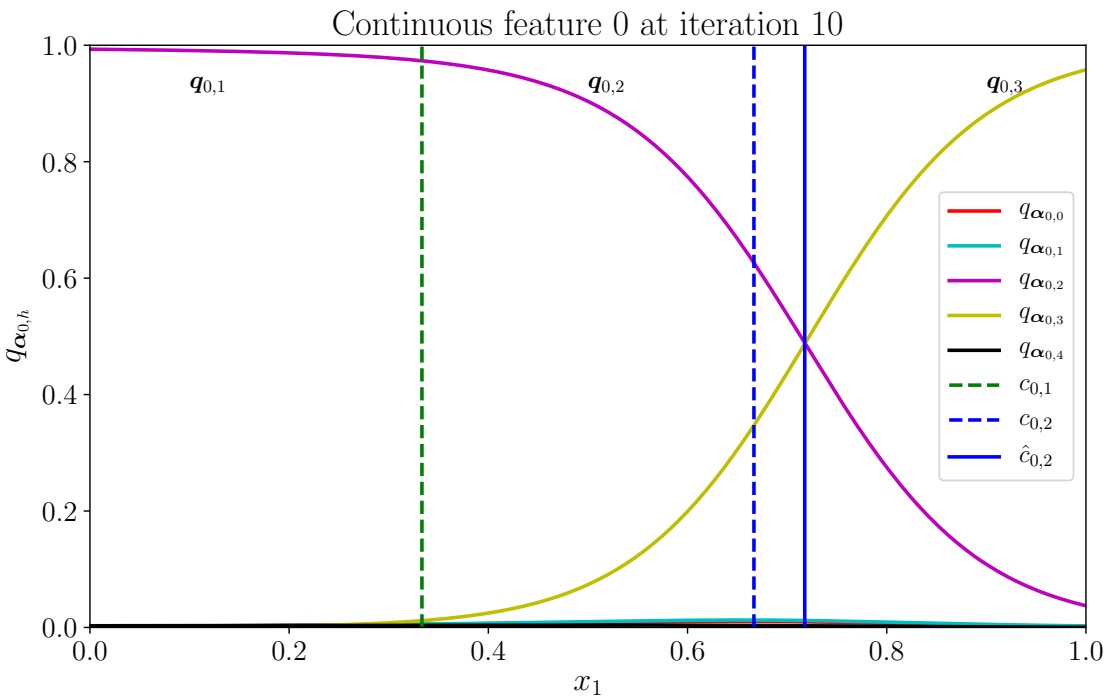

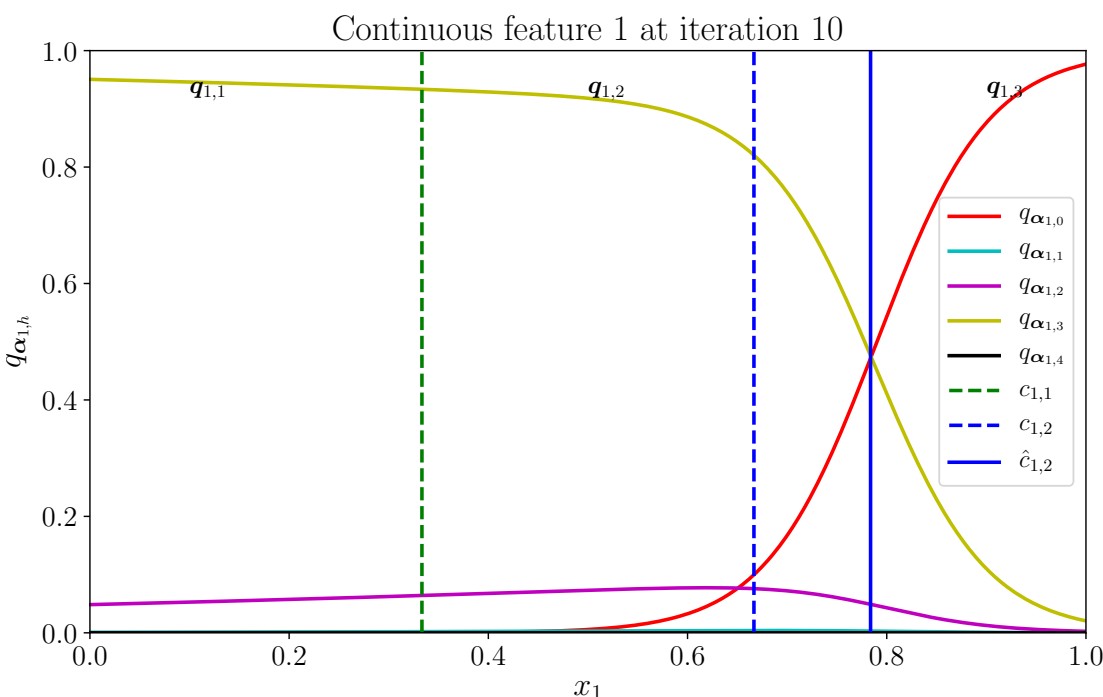

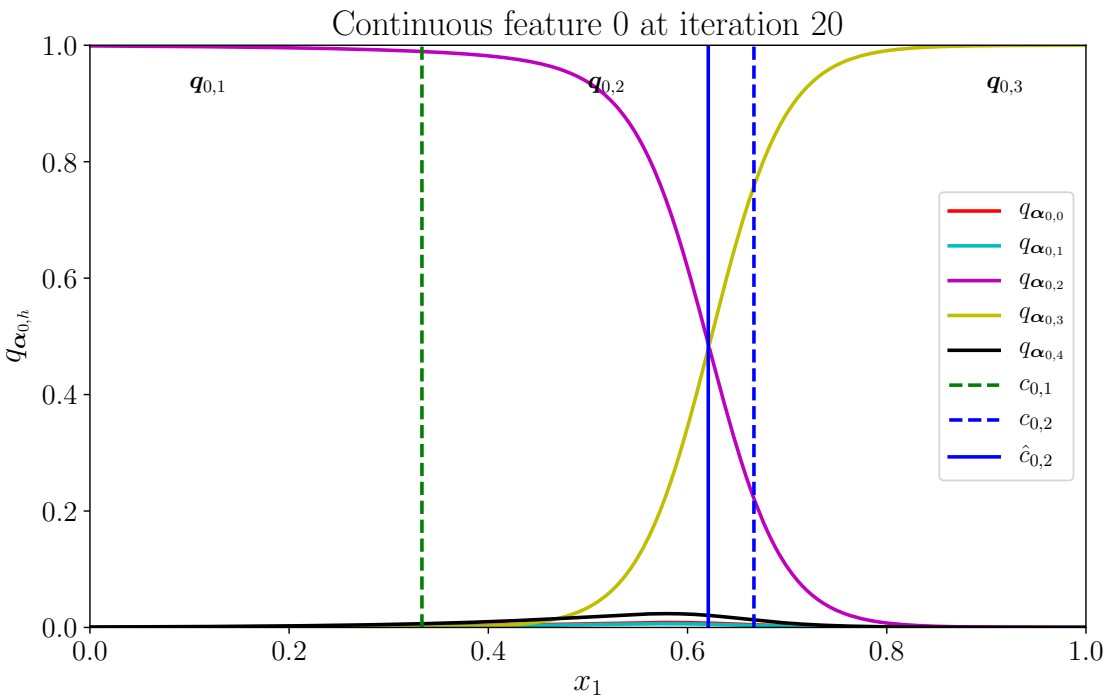

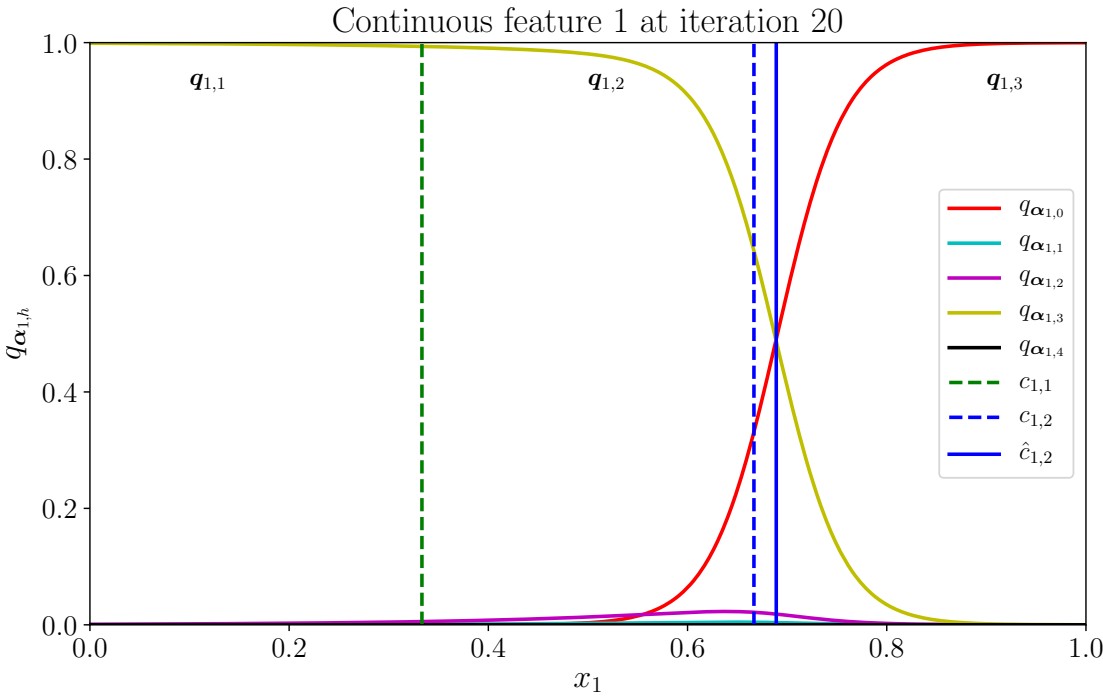

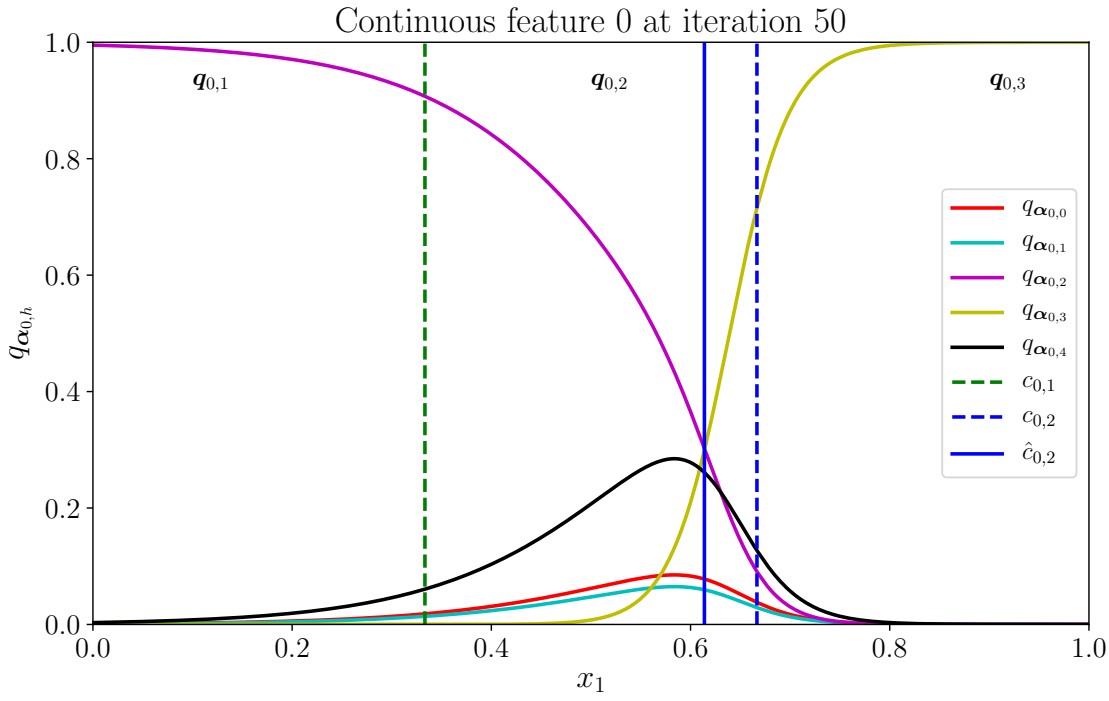

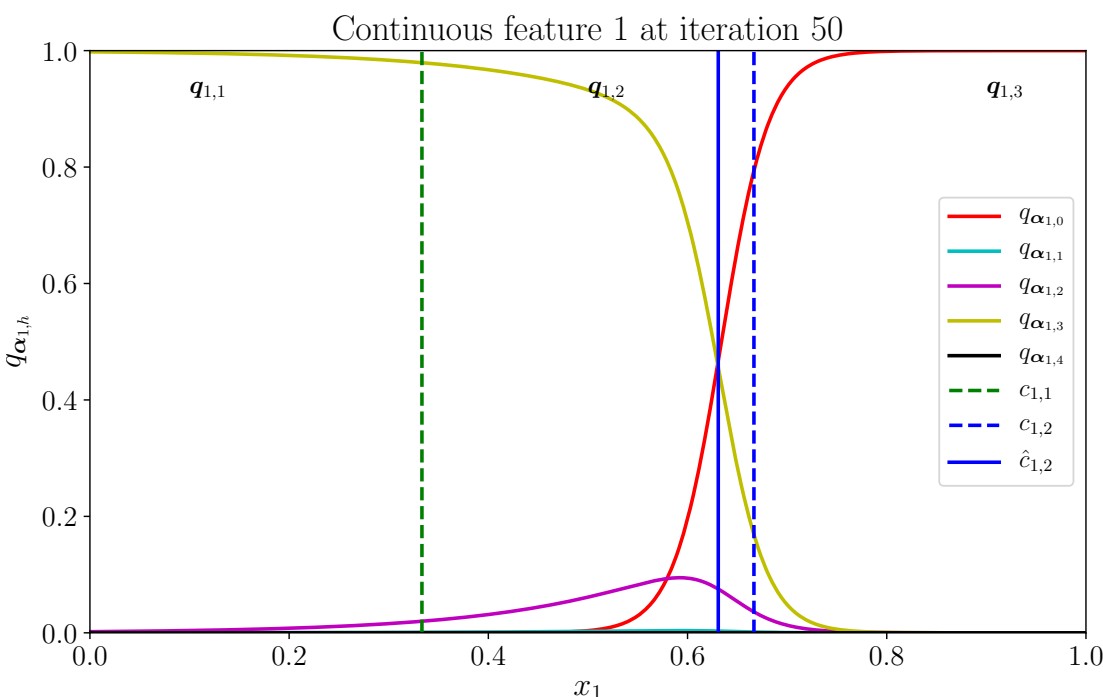

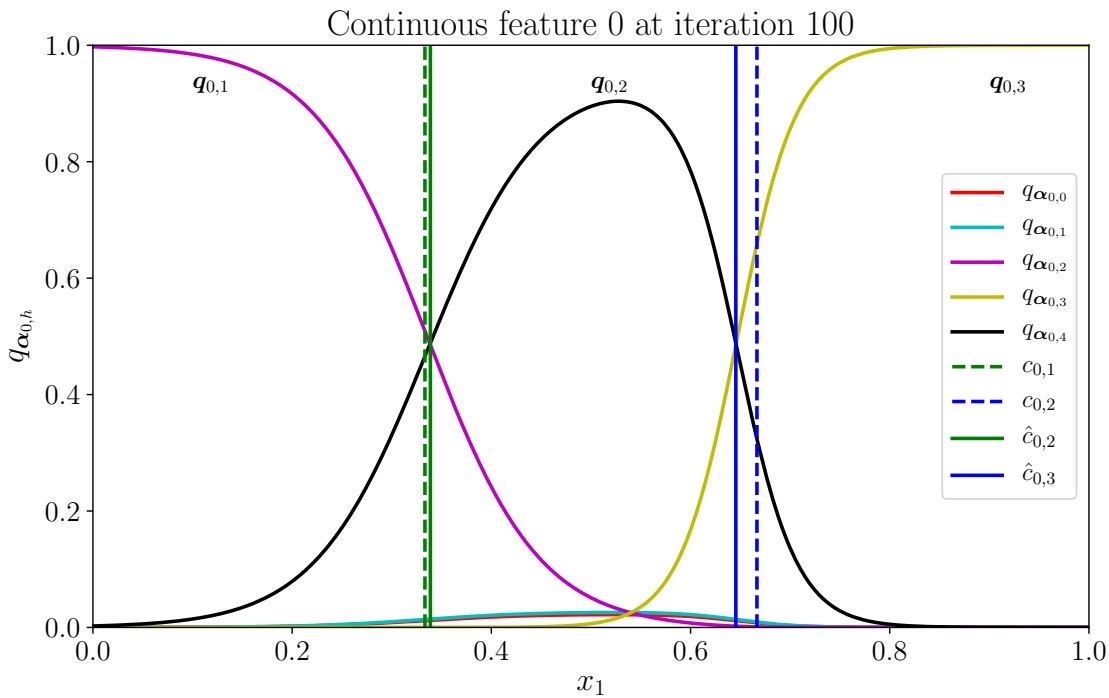

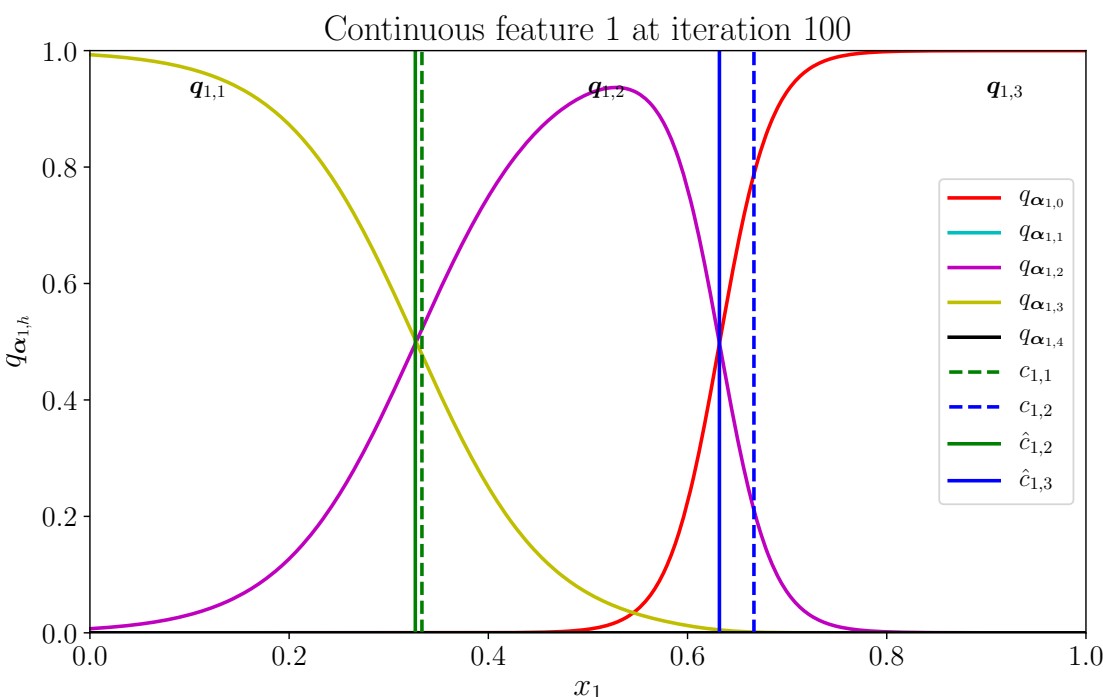

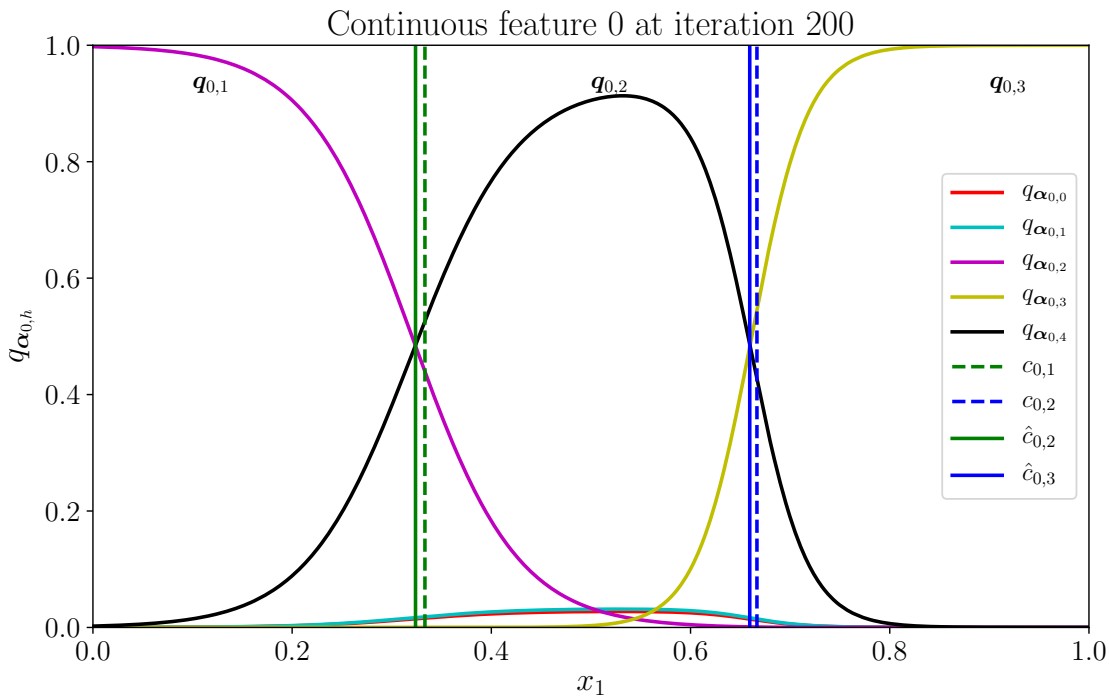

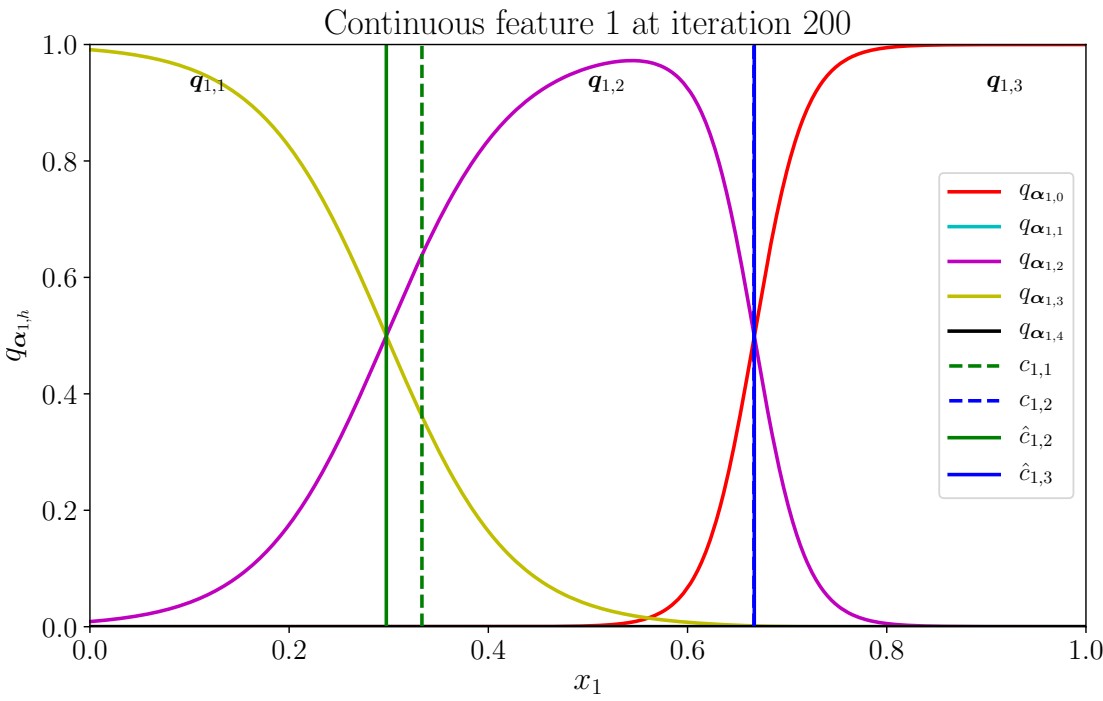

N.B.1: In this quick example, all inputs are in [0:1] so we did not perform any (batch) normalization.

N.B.2: The callback function could be improved, for example by not "hardcoding" x_qual and x_quant but getting either the training sample or the validation sample (if any) from the keras model. A possible solution to this **here**.

N.B.3: A lot of "tweaks" stemming from the deep learning community regarding optimization and bias / variance trade-off (e.g. Early Stopping rules, refined versions of stochastic gradient descent, ...) can improve the result but are out of the scope of this notebook.

N.B.4: Quantizations are given up to a permutation on the index $h$; for example, here for feature 1, $q_{\alpha,1,1}$ and $q_{\alpha,1,4}$ are "queezed" to 0.

**We were able to approximately recover the true discretization / grouping mechanism $q^\star$ that is the representation of the original data that yields the "best" logistic regression.**

