# OpenReview forum: "Feature quantization for parsimonious and interpretable predictive models"
_ICLR.cc/2019/Conference_

### Official Review · AnonReviewer2 · 2018-10-28
**many typos**

**Rating:** 4
**Confidence:** 2

**Review:**

i take reviewing very seriously, and it often takes hours per paper. this paper, however, has many typos, grammatical errors, and seems to have been submitted last minute.  therefore, i have read the paper quickly.
that said, i do not understand the results.
clearly, many discretization methods have previously been described, as alluded to by citing the taxonomy paper on the subject.  the authors state they have developed a better approach.  however, i do not see a comparison to the state of the art in the simulations, and i do not follow the results of Table 2, which columns correspond to which particular algorithms? in either case, the proposed approach does not seem to improve the empirical results, nor have theoretical guarantees, so i am not particularly impressed with the results either.

---

> ### Author Response · Authors · 2018-11-23
> **The paper had typos and was updated accordingly. Comparison to the state of the art should appear more clearly.**
>
> Thank you for your review. We are aware that the first submitted version has typos and mistakes and have updated the paper accordingly. We apologize for the inconvenience and hope you might see the revised paper with new eyes. Moreover, we have addressed your following remarks:
>
> - The state of the art consists in three baselines in Table 2 and 3:
> * A standard logistic regression (first column)
> * The current performance (‘manual’ in-house approach - second column)
> * Ad hoc methods from the literature (MDLP (Fayyad & Irani (1993)) and Chi2 independence tests to group factor levels - third column)
>
> We developed the literature review in Section 2.1 and in the experiments on real data in Section 4.2 to clarify this point.
> As shown in Table 2 and 3 it outperforms the first two baselines on all datasets, performs better than ad hoc methods on UCI, and on par / sometimes worse on Credit Scoring data.
>
> Although we do not provide theoretical guarantees, experiments on simulated data (Table 1) seem to show consistency of the proposed approach (see the figures in the Appendix).

---

### Official Review · AnonReviewer3 · 2018-10-29
**The presentation is unclear**

**Rating:** 3
**Confidence:** 3

**Review:**

This paper presents a feature quantization technique for logistic regression, which has already been a common practice in
 many finance applications.  The text feels rushed. From the current presentation, I find it difficult to understand what is the motivation of adopting the proposed relaxation of the optimization method, and how is the neural network-based estimation strategy connected to the logistic regression model. It seems the difference lies in the parameterized nonlinear transformation such that the cutting points can be somehow optimized.  The quality of the experiments performed is way below the expectation for ICLR. Although numerical experiments are performed on both simulated data and credit scoring data, it is still unclear whether the proposed method has superiority over competitors.

Question: In the test phase, how would the proposed method handle features that are not seen in the training phase?

---

> ### Author Response · Authors · 2018-11-23
> **The paper was admittedly unclear and was updated accordingly. Experiments have been enhanced.**
>
> Thank you for your review. We are aware that the first submitted version has typos and mistakes and have updated the paper accordingly. We apologize for the inconvenience and hope you might see the revised paper with new eyes. Moreover, we have addressed your following remarks:
>
> - The motivation behind the relaxation is that it can approximate the discrete quantization functions as now shown on Figure 2. The parameters of this relaxation are easily obtained via a very simple neural network, contrary to a greedy intractable search of the best quantization functions as in Equation (6). In a way, we use the proposed neural network as a computational graph to get the best quantization through the optimization of the \alpha parameters and the use of fast and standard libraries for deep learning.
>
> - Regarding the experiments, the simulated data are here only to to show empirically the consistency of the proposed method in various settings. As for real data, we understand the number of datasets on which we compare our proposed method to standard methods was not enough, so that we added 3 portfolios from Crédit Agricole Consumer Finance as well as 6 datasets from the UCI library, some of which also in the field of Credit Scoring.
>
> In your question, we suppose that by “features”, you meant categorical factor levels (say we have a categorical features with 5 levels of which only the 4 ‘first’ are seen during training). If our interpretation of your question is correct, recall that the neural network is only a proxy: in the end, our method yields a logistic regression, so that we need to estimate a coefficient for each factor level (as in Equation (4)) and p(y|x=5) cannot be calculated. This is the case for any “standard” supervised classification method to our knowledge. Missing values in categorical or continuous features can be addressed if there was missing values in these features in the training phase, thus forming a level labelled as “missing” that can be grouped with other levels / discretization intervals.

---

### Official Review · AnonReviewer1 · 2018-11-02
**The paper does not solve the question it proposed**

**Rating:** 2
**Confidence:** 4

**Review:**

The paper describes a question about discretize continuous features or group discrete features in the preprocessing step, which they call feature quantification. It considers that a joint training of feature quantification and a discriminative model can lead to a better performance than treating feature quantification as a preprocessing step.

This paper has many typos, grammar mistakes and question marks, which make it hard to follow. The question proposed is simple and easy to understand. However, I don't convinced by the solution in this paper. Since it is a hard optimization question, the authors proposed a relaxation approach in section 3.1. I do not think that exp(a+bx) is able to approach step functions since exp(a+bx) is monotone. I think Figure 1 is misleading. For grouping discrete features, the author propose to use exp(\alpha_{x_j, j}^h) and hoping that some \alpha parameters can be optimized to be equal, which is too simple. The exponential transformation here does not have an effect. It is more interesting to consider how to add some constraints. For example, if the discrete feature is ordinal, how one can assure that the grouped discrete feature is still ordinal. The relaxation in this paper is too much without handling any interesting constraints and the proposed exp(a+bx) can not approach step functions. The authors do not provide a good way to select number of cut points, which I think is a hard but interesting question.

The work also lacks value in literature review, optimization and experiments.

---

> ### Author Response · Authors · 2018-11-23
> **It does - softmax functions can approximate gate functions and provide an automatic way of selecting the number of cutpoints**
>
> Thank you for your review. We are aware that the first submitted version has typos and mistakes and have updated the paper accordingly. We apologize for the inconvenience and hope you might see the revised paper with new eyes. Moreover, we have addressed your following remarks:
>
> - The proposed relaxation is proportional to exp(a+bx) through, as was/is stated in Section 3.2, a softmax layer. As a consequence, the normalization also depends on x. For example, a discretization in 3 levels would be relaxed as (exp(a_1 + b_1 x) / sum_{h=1}^3 exp(a_h + b_h x), exp(a_2 + b_2 x) / sum_{h=1}^3 exp(a_h + b_h x), exp(a_3 + b_3 x) / sum_{h=1}^3 exp(a_h + b_h x)) that sum to 1 for each observation x. We updated the paper (Section 3.1) so that this normalization should now clearly appear.
>
> - Figure 1 and Figure 2 have been swapped for clarity. The Figure that you refer to as misleading, now Figure 2, has been replaced by the actual best discretization chosen by the proposed approach in experiment (a), where we see that the proposed relaxation is able to approximate the true data generating mechanism (i.e. the true quantization gate functions). In the appendix, we illustrate how, after some epochs, the proposed relaxation is able to converge to the gate functions and, in experiment (b), how it is able to “shut off” 2 neurons among 5 over the training data, thus approximating again the true data generating mechanism although we provided it with much more capacity than needed.
>
> - For grouping levels of categorical features, as you have rightfully stated, the exponential has no effect. It is here only to emphasize the straightforward equivalence of this parametrization and a softmax layer. As a consequence of this normalization, q_alpha can be interpreted as a probability of each of the (possibly numerous) factor levels of belonging to each group.
>
> - Concerning ordinal features, it is not in this paper’s scope. Moreover, in the logistic regression setting, it is actually much easier (because much less combinatorial) to solve by e.g. a Fused-Lasso penalization between all pairs of consecutive factor levels.
>
> - Concerning the selection of the number of cutpoints, as was stated in Section 3.2 (now Section 3.1) and in particular in Equation (10), it is done automatically by starting with a high number m_j of hidden neurons per feature which correspond to the maximum number of cutpoints plus one. Being able, by learning appropriate weights alpha, to “shut off” some neurons, the proposed neural network can explore quantizations \hat{q}_j from 1 to m bins, optimizing indirectly in each case the location of the cutpoints. This phenomenon is/was illustrated on Table 2, experiment (b), and is now as well illustrated in the appendix, as stated above.
>
> - Concerning the literature review, we added some comments about the already cited paper (Ramirez-Gallego et al. (2016)), in Section 2.2 (now Section 2.1). Moreover, we developed the references to two procedures in Section 4, namely MDLP and Chi2 tests of independence for grouping factor levels.
>
> - Concerning “optimization”, we guess you might refer to the gradient descent algorithm(s) and its / their hyper-parameters. In the first version of the paper, we used standard SGD with standard hyperparameters from Keras (lr=0.01, momentum=0.0, decay=0.0, nesterov=False). In this second version, we rely on RMSProp with a higher than default learning rate (lr=0.5, rho=0.9, epsilon=None, decay=0.0), as can be seen in the Appendix. Way better results could probably be obtained by changing the optimizer and / or its hyperparameters.
>
> - Concerning experiments, simulated data are here only to show empirically the consistency of the proposed method in various settings. As for real data, we understand the number of datasets on which we compare our proposed method to standard methods was not enough, so that we added 3 portfolios from Crédit Agricole Consumer Finance as well as 6 datasets from the UCI library, some of which also in the field of Credit Scoring.

---

### Author Response · Authors · 2018-11-23
**Revised version - short "cover letter"**

A revised version of the paper has been uploaded which, independently from the reviews, aims at:

- Correcting the typos present in the first version.

- Changing the notation of the quantization functions from f_j to q_j (which stands for quantization).

- Changing the result of the quantization, which was previously a single attribute with integer levels, for a dummy-encoding scheme (see Equation 1).

- Re-running the experiments on the simulated data, following the fact that we found RMSprop to work better than standard SGD.

Following the reviews, we also incorporated the following modifications:

- Adding several real data experiments by the addition of 3 portfolios of Crédit Agricole Consumer Finance (Table 3) and 6 datasets from the UCI library (Table 2).

- Enriching the literature review by adding some detail about the already cited taxonomy (Ramirez-Gallego et al. (2016)).

- Providing some detail on the ad hoc methods used in Table 2 (MDLP (Fayyad & Irani (1993)) and Chi2 independence tests to group factor levels).

- Clarifying Figure 1 (now Figure 2) to show that the smooth approximation can indeed approximate a “hard” quantization.

- Adding a Jupyter Notebook in the Appendix to witness the empirical consistence of the proposed approach.

The first version of the paper was admittedly hard to read due to typos, diverse mistakes and some missing references. We misunderstood the submission deadline hour. Nevertheless, we hope to be reread with new eyes as we think our proposal can be impactful for practitioners relying on discretization and grouping of factor levels, as we witness in the Credit Scoring industry where it is already successfully used by Crédit Agricole Consumer Finance.

---

### Meta-Review · Area_Chair1 · 2018-12-14
**Meta-Review for interpretable predictive models paper**

**Confidence:** 5
**Recommendation:** Reject

**Metareview:**

All reviewers agree to reject. While there were many positive points to this work, reviewers believed that it was not yet ready for acceptance.